# Role of Microparticles in the Pathogenesis of Inflammatory Joint Diseases

**DOI:** 10.3390/ijms20215453

**Published:** 2019-11-01

**Authors:** Magdalena Krajewska-Włodarczyk, Agnieszka Owczarczyk-Saczonek, Zbigniew Żuber, Maja Wojtkiewicz, Joanna Wojtkiewicz

**Affiliations:** 1Department of Rheumatology, Municipal Hospital in Olsztyn, 10-900 Olsztyn, Poland; 2Department of Internal Medicine, School of Medicine, Collegium Medicum, University of Warmia and Mazury, 10-900 Olsztyn, Poland; 3Department of Dermatology, Sexually Transmitted Diseases and Clinical Immunology, School of Medicine, Collegium Medicum, University of Warmia and Mazury, 10-900 Olsztyn, Poland; aganek@wp.pl; 4Department of Pediatrics, Faculty of Medicine and Health Sciences, Andrzej Frycz Modrzewski Kraków University, 30-705 Kraków, Poland; zbyszekzuber@interia.pl; 5Faculty of Earth Sciences, Department of Geomatics and Cartography Nicolaus Copernicus University, 87-100 Torun, Poland; maja.wojtkiewicz@umk.pl; 6Department of Pathophysiology, School of Medicine, Collegium Medicum, University of Warmia and Mazury, 10-900 Olsztyn, Poland; joanna.wojtkiewicz@uwm.edu.pl

**Keywords:** microparticles, joint inflammatory diseases

## Abstract

Rheumatoid arthritis (RA), juvenile idiopathic arthritis (JIA), ankylosing spondylitis (AS), and psoriatic arthritis (PsA) make up a group of chronic immune-mediated inflammatory diseases (IMIDs). The course of these diseases involves chronic inflammation of joints and enthesopathies, which can result in joint damage and disability. Microparticles (MPs) are a group of small spherical membranous vesicles. The structure and cellular origin of MPs, mechanisms that stimulate their secretion and the place of their production, determine their biological properties, which could become manifest in the pathogenesis of immune-mediated inflammatory diseases. Microparticles can stimulate synovitis with proinflammatory cytokines and chemokines. MPs may also contribute to the pathogenesis of rheumatic diseases by the formation of immune complexes and complement activation, pro-coagulation activity, activation of vascular endothelium cells, and stimulation of metalloproteinase production. It seems that in the future, microparticles can become a modern marker of disease activity, a response to treatment, and, possibly, they can be used in the prognosis of the course of arthritis. The knowledge of the complexity of MPs biology remains incomplete and it requires further comprehensive studies to explain how they affect the development of rheumatic diseases. This review focuses on the immunopathogenic and therapeutic role of MPs in chronic immune-mediated inflammatory joint diseases.

## 1. Introduction

Cell membrane microparticles (MPs), or microvesicles, are fragments of surface membranes of activated eukaryotic cells. Their size, which determines their diameter as lying within the interval of 0.1 to 1 μm, is their main defining criterion. Therefore, the diameter of MPs is greater than that of exosomes and smaller than that of apoptotic bodies or small platelets. In physiological conditions, when cells mature, age, and undergo apoptosis, microparticles are released by exfoliation or by shedding to body fluids from cell membranes of all morphotic elements of blood and vascular endothelium [1,2]. MPs can be found in plasma, in whole blood, in umbilical blood, in cerebrospinal fluid, in urine, in milk, and in saliva. Microparticles do not have a cell nucleus, but they contain cytoplasmic material and surface antigens of their parent cells, owing to which their origin can be determined [2,3,4] (Table 1). Increased secretion of MPs in physiological conditions takes place in pregnant women, after intensive physical effort, in obese people, and in smokers [5]. Increased secretion of microparticles from activated platelets, leukocytes, erythrocytes, smooth muscle cells, and vascular endothelium cells takes place in immune-mediated diseases. An increased number of microparticles have been found in immune thrombocytopenia [6], in systemic lupus erythematosus [7], in rheumatoid arthritis [8], and in psoriasis [9,10]. The presence in MPs membrane of intercellular adhesion molecule 1 (ICAM-1) and vascular cell adhesion molecule 1 (VCAM-1) enables microparticles to join other cells and to take part in intermembrane transport of enzymes and receptor proteins, cytokines, growth factors, and nucleic acids: Micro RNA (miRNA), messenger RNA (mRNA), and deoxyribonucleic acid (DNA) [11,12]. 

As many as 90% of all circulating microparticles are MPs derived from platelets and megakaryocytes (PMPs) [13]. PMPs have a number of receptors on their membrane surface, including adhesive proteins. For PMPs, the most frequent surface markers are: Glycoprotein IIb (CD41), Ib (CD42b), IIb/IIIa (CD41a), IIIa (CD61), selectin P (CD62P) [3], and sphingolysine, arachidonic acid (AA), and bioactive lipids [5,14,15]. Contact of platelet-derived microparticles with target cells can result in monocyte chemotaxis, stimulation of cytokine secretion, activation of endothelial cells, and increased tissue factor expression on endothelial cell surface [16]. Platelet microparticles stimulate phagocytic activity of granulocytes by increasing the expression of the adhesive molecule CD11b on them [17]. An increased number of platelet-derived microparticles have been observed in atherosclerosis [18], diabetes [19], coronary artery disease [20], thrombotic thrombocytopenic purpura [21], aplastic anaemia, and paroxysmal nocturnal haemoglobinuria [22]. However, it is very likely that the activation of monocytes/macrophages, B-cells, T-cells, and endothelial cells observed in patients with inflammatory diseases may result in an increased release of MPs from these cells, raising their levels in plasma.

It has been proposed that excessive production of MPs may predispose to autoimmune diseases such as rheumatoid arthritis and systemic lupus erythematosus [23] but their role in the pathogenesis of these autoimmune diseases may differ. In patients with SLE, a prototypic autoimmune disease characterized by the production of antibodies to components of the cell nucleus and the formation of immune complexes, circulating MPs differ in their amount and composition compared to from those in patients with RA or healthy controls. MPs from SLE patients contain more immunoglobulins (IgG, IgA, and IgM) and complement components (C1q, C1s, C3, C4b, and C9) on their surface indicating the role of MPs as a source of immune complexes [24]. MPs containing DNA and RNA in SLE can behave as self-adjuvants for the production of autoantibodies. They can also increase tolerance of immature B-lymphocytes and break the tolerance of mature B-cells. MPs endocytosed by plasmacytoid dendritic cells are able to contact intracellular TLR7 (toll-like receptor-7) and TLR9, leading to the production of proinflammatory cytokines including type 1 interferon (IFN-1) and IL-6 (interleukin-6) [25].

## 2. The Mechanism of Microparticle Formation

The primary settings for MPs release are cellular activation and death. An increase in the intracellular concentration of calcium ions secreted by cytoplasmic reticulum is a response to multiple factors, including: An increase in the number of free radicals, increased shear force, adenosine diphosphate (ADP) secreted by activated platelets, expression of CD40 ligand (CD40L) on T-cells [26]. In these conditions, activation of calcium concentration-dependent enzymes takes place; these include gelsolin, which facilitates the separation of actin fibers from platelet cytoskeleton [27]; aminophospholipid translocase, which transports aminophospholipids from the outer membrane into the cell interior [28]; floppase, which transports phospholipids from the inner lipid bilayer outwards [29]; calpain, which destroys cytoskeleton actin fibers [30]; and scramblase, which affects transmembrane phospholipid transport [31] (Figure 1). The activated enzymes contribute to the loss of asymmetric distribution of phospholipids in the cell membrane in which phosphatidylserine (PS) and phosphatidylethanolamine (PE) are present mainly in the inner cytoplasmic layer, and phosphatidylcholine (PC) and sphingomyelin (SM) are present in the outer layer of the lipid bilayer. The asymmetry loss process resulting from transferring phosphatidylserine and phosphatidylethanolamine to the outer layer of the cell membrane and simultaneous cytoskeleton destabilization allow the formation and secretion of microparticles [26,32].

An increase in the intracellular concentration of calcium, as a response to cell activation or apoptotic stimuli, results in the activation of calcium concentration-dependent enzymes: Gelsolin, aminophospholipid translocase, floppase, calpain, and scramblase. During this process, membrane asymmetry is lost, leading to the exposure of phosphatidylserine and phosphatidylethanolamine normally present in the inner leaflet of the membrane bilayer. Destabilization of the cytoskeleton results in cellular contraction and membrane blebbing.

## 3. Methods of Microparticle Detection

There are no standardized microparticle testing techniques and each laboratory conducting such testing develops its own MPs detection methods and standards. However, common protocol among researchers is to start collecting MPs from blood with a centrifugation to collect platelet-free plasma to avoid the activation and subsequent release of microparticles from platelets. Microparticle identification is a technical challenge because they are much smaller than cells of origin (diameters of 10–100× less). Microparticles can be isolated from blood, other biological fluids, or from cell cultures. Since MPs are cell-derived structures, a cytometric test is the “gold standard” and is the most widely used method of microparticle detection on the basis of light scattering as well as binding of marker antibodies to identify the cell of origin [33]. The MPs population to the flow cytometry settings is defined by using size calibration beads. However, MPs smaller than approximately 0.5 µm in diameter are not efficiently resolved by conventional flow cytometers. Available digital flow cytometers do not count all MPs because of their limited forward scatter (FS) sensitivity [34]. Recently, high-sensitivity flow cytometers with significantly improved light scatter detection became available and provide sufficient size resolution for the identification of MPs subtypes [35]. Microparticles generally expose phosphatidylserine and membrane antigens of their parent cells. MPs as cell-derived membrane structures in the cytometry test are commonly determined with annexin V, usually stained with phycoerythrin and antibodies against specific cell antigens, stained with fluorescein isothiocyanate. Labeled annexin V binds mainly to phosphatidylserine, which is typical of all types of microparticles and which is present on the MPs membranous surface and less frequently in membrane permeability disorders; it can also connect to phosphatidylserine located inside microparticles [36]. Annexin V is often used to identify microparticles, but some MPs may not show a tendency to bind to this protein [37]. The use of antibodies against different membranous antigens enables identification of the cellular origin of microparticles. 

Another method of microparticle determination is based on ELISA (enzyme-linked immunosorbent assay), which makes use of test plates coated with annexin V or antibodies specific to cell membrane antigens [38]. When total phosphorus or phospholipid activity is determined, ELISA enables quantitative determination of microparticles [39]. 

## 4. Role of Microparticles in Inflammatory Joint Diseases

Rheumatoid arthritis (RA), juvenile idiopathic arthritis (JIA), ankylosing spondylitis (AS), and psoriatic arthritis (PsA) are chronic immune-mediated inflammations, leading to chronic joint inflammations and/or enthesopathies and to many extra-articular complications. Increasing numbers of circulating microparticles in immune-mediated diseases have been reported; the increase is particularly manifested if vessels are also affected; it usually concerns microparticles of platelet origin and—less frequently—those of endothelial origin [40,41,42,43,44,45,46]. The few studies conducted to date have suggested, or even confirmed, a pathogenic link between microparticles and immune-dependent diseases [6,7,8,9,10,47,48]. MPs can be detected in inflammatory joint diseases in blood and other biological fluids (Table 2).

### 4.1. Rheumatoid Arthritis

An increase in circulating MPs secretion is closely linked to an increase in cytokine production and appears to be a significant factor which affects inflammation development within the synovial membrane in rheumatoid arthritis [46]. In one of the first studies assessing the relationship between MPs and rheumatoid arthritis in a group of 19 patients, Knifjj-Dutmer et al. observed an increased number of circulating PMPs compared to a group of healthy individuals, and a significant relationship between the number of circulating microparticles of platelet origin and the disease activity assessed by the DAS28 score [49]. Moreover, researchers suggested a possible effect of PMPs on the development of cardiovascular diseases in patients with RA, leading to increased mortality linked to vascular complications, compared to the general population [49]. Viñuela-Berni et al. observed an increased number of MPs with CD3, CD14, CD19, CD41, and CD63E antigens in plasma of RA patients with high disease activity (DAS 28 > 5.1) [52]. The link between the intensity of inflammation in RA and the number and activity of released microparticles of endothelial origin was confirmed by Barbati et al. The total pool of circulating MPs and endothelial MPs (EMPs) initially increased, then decreased after four months of anti-TNFα therapy [58]. The microparticles with CD3, CD14, CD19, CD41, and CD63E antigens also stimulated secretion of TNFα and IL-1, IL-17 by monocytes in vitro [52]. The potential role of microparticles in the pathogenesis of rheumatoid arthritis appears to be very complex (Table 3). 

Apart from serum, microparticles have been detected in other body fluids of the patients. Viñuela-Berni et al. observed an increased number of MPs with CD3, CD14, and CD19 antigens in the urine of RA patients with high disease activity [52]. Boilard et al. [8] analyzed samples of synovial fluid in RA patients and found it to contain large numbers of PMPs (slightly less than 2 × 105 CD41+ MPs/µL). Synovial fluid of RA patients contained (much less than PMPs) MPs with surface antigens of neutrophils, monocytes, and T-cells. It was an interesting observation to determine a group of neutrophils present in rheumatoid synovial fluid, with simultaneously present leukocyte CD45 antigen and a platelet CD41 antigen. The immunofluorescence signal was a result of attaching to neutrophils of the bodies corresponding to PMPs rather than whole platelets [8]. Interestingly, the number of PMPs determined in rheumatoid fluid in this study was much higher than in serum of RA patients in the study by Knijff et al. [49], where it was 600 per µL. Michael et al. [54] determined the number of MPs in synovial fluid in RA patients and found a considerably larger number of MPs derived from granulocytes, monocytes, and T-cells than in patients with degenerative joint disease and in the control group. Moreover, the number of granulocyte-derived MPs in RA patients was significantly larger in the patients with aCCP antibodies. 

It is not completely clear how platelets infiltrate synovial fluid, although undamaged platelets, platelet aggregates, and platelets attached to leukocytes have been found in studies with RA patients for many years [66,67,68]. It is probable that collagen, fibrinogen, proteolytic enzymes, inflammatory cytokines, and shear forces in a joint could stimulate the production of platelet-derived MPs. It is also possible that, owing to their size, microparticles can penetrate synovial fluid and synovial membranes because the number of platelet microparticles in synovial fluid in RA patients is considerably larger than the number of PMPs in peripheral blood, which may suggest locally increased microparticle release from activated platelets in vessels near joints. Platelets can be activated locally by stimulation through collagen, a specific platelet receptor containing glycoprotein VI (GPVI) [69]. In a study by Boilard et al., platelet activation mediated by the GPVI, conducted in vivo on an animal model with transgenic K/BxN mice, induced release of microparticles containing both forms of interleukin 1 (IL-1): IL-1α and IL-β, stimulating production and secretion of IL-6 and IL-8 by fibroblast-like synoviocytes (FLS). The findings of the study suggest the pro-inflammatory potential of PMPs and their active participation in pathogenesis of rheumatoid arthritis [8]. Activation of platelet receptor, GPVI, leads to activation of spleen tyrosine kinase (SYK) in platelets and B-cells, further activating Bruton’s tyrosine kinase (BTK), which plays a key role in the activation of B-cells, which is essential in their proper function and development. Bruton’s kinase inhibition with the BTK selective inhibitor was examined by Hsu et al. and found to reduce collagen-induced production of PMPs [70]. In another study, BTK blockade in activated platelet culture also resulted in a decrease in production of microparticles and inhibition of production and release of IL-6 and IL-8 [71,72]. 

Apart from platelet-derived microparticles, synovial fluid in RA patients was found to contain MPs from monocytes, granulocytes, T- and B-cells, and erythrocytes [50,73,74]. Monocyte-derived MPs, as determined by Headland et al., were present in a much larger amount in synovial fluid than in plasma of RA patients. Researchers obtained some interesting findings on an animal model, where they observed a protective effect of monocyte-derived microparticles on articular cartilage, which is associated with decreasing secretion of IL-8 and prostaglandin E2 [53]. 

Microparticles exhibit high pro-coagulation activity mediated by the TF/VII factor, thereby stimulating thrombin production. They can also contribute to developing proatherogenic vasculitis and to the formation of “rice bodies” within joints as a manifestation of local coagulation processes [64]. In their study with RA patients, concerning non-differentiated arthritis, Berckmans et al. found incubation of MPs obtained from synovial fluid in the presence of FLS obtained by the biopsy of the synovial membrane to result in an increase in production and release by synoviocytes of IL-6, IL-8, monocyte chemoattractant protein 1 (MPC-1), RANTES (regulation on activation normal T-cells expressed and secreted) chemokine, and vascular endothelium growth factor (VEGF) [63]. A local increase in VEGF secretion can stimulate angiogenesis within articular tissues in RA, especially at its early stages [74]. Stimulation of angiogenesis in joints affected by RA can also be mediated by chemokines. Reich et al. observed a stimulating effect of leukocyte-derived MPs on production and release by rheumatoid synoviocytes of proangiogenic CXC chemokine with an ELR motive—a sequence of three amino acids: Glutamic acid-leucine-arginine (Glu-Leu-Arg). In their study, the authors observed increasing mRNA expression for ligands of chemokine CXC ELR+: CXCL1, CXCL2, CXCL3, CXCL5, and CXCL6 [65]. 

Matrix metalloproteinases (MMP) are responsible for processes of extracellular matrix component transformation and degradation. Distler et al. demonstrated that MPs derived from B-cells and monocytes can stimulate rheumatoid FLS additionally to synthesis of metalloproteinases 1, 3, 9, and 13, which participate in the destruction of the extracellular matrix of cartilaginous and bone tissue in RA. In this study, the microparticles under study stimulated the production of IL-6, IL-8, MCP-1, and MCP-2 by fibroblasts [62]. 

The complement system, comprising approximately 40 proteins, plays an important role in the regulation of innate immune response by stimulating phagocytosis and intensification of an inflammatory reaction. The system is activated in a cascade manner. Removing apoptotic and necrotic cells is one of the functions of the complement system [75]. These cells activate the complement system mainly through a conventional pathway, as a result of a reaction of the C1q component with the CH2 domain of the antibody Fc fragment [76,77,78]. Microparticles with membranous features of apoptotic and necrotic cells (with exposed phosphatidylserine, phosphatidylethynolamine, oxidated phospholipids) can participate in the development of inflammation in RA by activating the complement cascade. It has been shown that MPs formed from apoptotic Jurkat leukemia cell [79] and stimulated neutrophils [80,81] can bind the C1q and, thereby, activate the complement in vitro through a conventional activation pathway. Biro et al. found C1q, C4, and/or C3 components on microparticles isolated from synovial fluid and—to a lesser extent—on microparticles obtained from serum [60]. It seems that microparticles, especially those derived from platelets, exposing the CD41 antigen, can also take part in the formation of immune complexes (IC), usually described as structures containing antibodies, antigens, and complement components. Cloutier et al. used high resolution flow cytometry and transmission electron microscopy and detected MPs in IC present in synovial fluid in RA patients, forming specific mpIC [61]. 

Microparticles in rheumatoid synovial fluid can undergo the process of protein citrullination—posttranslational deimination of arginine residues catalyzed by peptidylarginine deiminase (PAD), which results in the production of antibodies against cyclic citrullinated peptides (aCCP). For platelet microparticles described in the study by Cloutier et al., with expression of the FcγRIIa receptor, mpIC were formed not by connecting antibodies to this receptor, but by binding MPs to citrullinated proteins, such as fibrinogen and vimentin. The researchers confirmed that platelet MPs can react with aCCP antibodies in a mechanism which follows citrullination of PMPs surface proteins and by binding microparticles and citrullinated proteins. The paper also describes a stimulating effect of platelet mpIC on the production of leukotrienes by neutrophils. The researchers made an interesting observation by comparing the amount of mpIC in synovial fluid collected from RA and PsA patients. Despite the presence of MPs and immunoglobulins in synovial fluid of PsA patients, the amount of detected mpIC was nearly twenty times smaller than in RA patients (2000 ± 900 mpICs/µL vs 39,400 ± 9400 mpICs/µL) [82]. Unlike the previous study, two other studies found no proof of any relationship between the amount of circulating microparticles and immune complexes containing MPs in RA patients and the indices conventionally used to assess the disease activity [49,82]. This may indicate a highly effective vascular and reticular-endothelial mechanism of IC elimination or local formation of mpIC in joints affected by inflammation.

A very important role in pathogenesis of RA is played by activated B-cells. Data from the study by Messer et al. suggest the role of microparticles in synovial fluid in inducing the release of B-cell activating factor (BAFF), thymic stromal lymphopoietin (TSLP), and antileukoprotease (SLPI) by fibroblast-like synoviocytes. MPs present in synovial fluid stimulated the secretion of BAFF to the same extent as IFN-γ used as a control. The effect was observed both among the RA patients—study participants—and in patients with degenerative joint disease, which indicated the activity of MPs in stimulating B-cells, regardless of the disease type. The main difference in this study was quantitative. The number of MPs in synovial fluid in RA patients was considerably larger than in individuals with no joint inflammation. The study assessed the ability of MPs from THP-1 monocytic–macrophagic cell line and from the CEM lymphocyte line to synthesize and release BAFF, TLSP, and SLPI by activated FLS. Microparticles of monocytic–macrophagic origin had high inflammatory activity, which indicates the important participation of monocytes in initiation of an inflammatory response. MPs from activated T-cells which, in turn, stimulate the secretion of IL-6 and IL-8, did not stimulate the release of BAFF, but of TLSP and SLPI by synoviocytes. Those same MPs, did not affect the release of BAFF, TLSP, or SLPI after being treated with actinomycin D, which suggested no effect of MPs from apoptotic T-cells on activation of B-cells [59].

### 4.2. Juvenile Idiopathic Arthritis

There are only a few papers on circulating MPs in other joint inflammations. In a recent study with 26 children with JIA, Kumar et al. found a greater number of PMPs in plasma of patients with the active disease compared to the patients with disease remission [55] despite the absence of a difference in the number of platelets. Additionally the number of PMPs in synovial fluid in JIA was close to the number of PMPs in serum of individuals with active disease. The researchers suggested the potential usability of PMPs determination as a sensitive indicator of JIA activity. As in the study by Knifjj-Dutmer et al. [49], the number of PMPs in patients with the active disease did not correlate with the number of circulating platelets, ESR and CRP. Boilard et al. [8] analyzed samples of synovial fluid in JIA patients and found an increased number of PMPs in synovial fluid of patients with JIA, whereas the number of PMPs was not determinable in synovial fluid of 95% of patients with osteoarthritis.

### 4.3. Ankylosing Spondylitis

In a study with 82 male AS patients and a group of healthy individuals, Sari et al. did not find any difference in the number of PMPs or EMPs in plasma between the groups under study. No differences were also found between the number of PMPs and EMPs in patients with high disease activity as defined by BASDAI > 4 (Bath Ankylosing Spondylitis Disease Activity Index) and its low activity [56]. This notwithstanding, a significant decrease in the number of MPMs and EMPs was observed during the anti-TNFα treatment (etanercept, infliximab, adalimumab) compared with a conventional therapy. This may indirectly indicate the role of MPs in AS pathogenesis and, because both the number of PMPs and EMPs increases in vascular endothelium disorders, a positive vascular effect of anti-TNFα treatment. In a study with AS patients, Bradley did not observe any differences in the number of MPs between them and the control group; in contrast, significantly higher expression of CD4, CD62, CD14, VCam1 and lower expression of CD41 and CD54 was observed in the MPs surface in the patients compared with healthy individuals as well as significantly more frequent positive immunofluorescence of AV-labeled MPs in the patients [57], which implies a relationship between different cellular origin and a mechanism leading to MPs formation (in this case—apoptosis) and AS development.

### 4.4. Psoriatic Arthritis

An increase in the number of circulating endothelial, platelet, and monocyte-derived MPs in psoriatic patients was also observed in a study conducted by Takeshita et al. [9]. In another paper, Papadavid et al. described a considerable increase in the number of PMPs correlating with an increase in the concentration of interleukin 12 (IL-12) and an increase in the disease activity as assessed by the PASI (Psoriasis Area Severity Index) [83]. In another study with patients with severe psoriasis (with or without psoriatic arthritis), Ho et al. observed a larger number of circulating PMPs and EMPs in patients compared with the control group [47]. The researchers did not observe any differences in the number of MPs between patients with psoriasis and those with psoriatic arthritis. Contrary to expectations, no difference was observed in the number of PMPs or EMPs before and after a three-month treatment with IL12/23 p40 subunit inhibitor, despite a significant clinical improvement measured with PASI. Increased number of PMPs in synovial fluid of patients with PsA compared to osteoarthritis was described by Boilard et al. [8].

## 5. Microparticles as an Indicator of Disease Activity

Microparticles have been attracting increasing attention as potential indicators of eukaryotic cell activation. They could provide valuable information on inflammatory processes in progress, disease activity, and the response to treatment as well as prognosis as a disease assessment indicator. 

Hsu et al. demonstrated that release of MPs from platelets activated with collagen decreased considerably after the activity of kinase BTK was inhibited [70]. Reduction of PMPs numbers in other studies with BTK inhibitor in platelet cultures was associated with a decrease in production and the release of inflammatory cytokines IL-6 and IL-8 [71,72]. A considerable increase in the number of platelet, monocyte, and lymphocyte-derived microparticles (CD3, CD19) has been observed in RA patients with high disease activity [52]. Rodrigez-Cario et al. examined the amount of circulating microparticles and their origin in 114 RA patients. The total number of MPs in platelet poor plasma was much higher in individuals with arthritis compared to a group of healthy individuals. The occurrence of different MPs subtypes in this study differed considerably in the RA group and was associated with the clinical course of joint inflammation: The amount of endothelial MPs was associated with the disease duration, the amount of granulocyte MPs was associated with the disease activity as assessed by DAS28, whereas the amount of monocyte-derived MPs was associated with the presence of the rheumatoid factor. The amount of MPs was also associated with the presence of traditional cardiovascular risk factors [59]. The findings of Cloutier et al. could indicate the possibility of using the studies of circulating mpICs to assess RA activity [60]. However, different conclusions were presented by van Eijk et al. based on a study with 24 patients with an early form of RA [51]. The disease activity was assessed based on the ESR, C-reactive protein (CRP) level, and DAS28 score. Moreover, the level of serum amyloid-P (SAP) and the amount of circulating MPs and MPs presenting the C1q complement component was determined in the patients. Nine patients were reassessed after an eight-week intensive treatment according to the COBRA (COmBination therapy in Rheumatoid Arthritis) regimen, which included a combined treatment with methotrexate, sulfasalazine, and prednisolone. As expected, ESR, CRP, and DAS28 improved as a result of the treatment; however, contrary to expectations, neither the amount of circulating MPs, nor MPs with the attached C1q component decreased, which suggests the absence of any connection between the activity of inflammation and MPs release and mpIC production [51]. In a recent paper, Chen demonstrated that inhibition of PMPs formation in an animal model of CIA (collagen-induced arthritis) and reduction of circulating PMPs was associated with a clinical decrease in the disease activity assessed as joint swelling and stiffness [84].

## 6. The Potential Role of Mesenchymal Stem Cells-Derived Microparticles in Inflammatory Joint Disease Therapy

Immunomodulating properties of mesenchymal stem cells (MSC) are used in current studies of new therapeutic options in inflammatory joint diseases [85,86]. Cosenza et al. studied the delayed-T hypersensitivity model (DTH) and CIA and found MSCs-derived MPs administered parenterally to have an immunosuppressive effect by inhibiting T- and B-cell proliferation and inducing Treg cells [87]. Compared to MSC, MSCs-derived MPs were more effective in stimulating CD4+CD25+Foxp3+ Treg and CD4+IL-10+ Tr1 in vitro. In the DTH model, MSCs-derived MPs proved to be more effective in inhibiting the inflammation than MSC, and they significantly inhibited the formation of erosions in the CIA model. In another paper, Cosenza et al. described the anti-inflammatory effect on macrophage maturation of MSCs-derived MPs with lower membranous expression of TNFα and higher expression of IL-10 [88]. Microparticles formed from adipose-derived MSCs (ASCs) can inhibit in vitro the expression of inflammatory cytokines and chemokines secreted by fibroblast-like synoviocytes [89] and also increase the production of anti-inflammatory IL-10 and collagen II in chondrocyte cultures [90].

## 7. Summary

Microparticles have special biological properties which allow them to play a role in pathogenesis of chronic inflammation. They can also be used as a sensitive indicator of an inflammation in progress. Most studies with patients with joint inflammations have reported an increase in circulating MPs and MPs in synovial fluid in joints affected by the inflammation. Participation of microparticles in the pathogenesis of RA, JIA, AS, and PsA is complex. Microparticles can stimulate the production and release of inflammatory factors, take part in their transport, in the formation of immune complexes, and induce formation of autoantibodies. In future, MPs determination can be used as one of the elements of disease activity assessment, of monitoring the response to treatment, or forecasting the course of a joint inflammation. Microparticles derived from stem cells can also become a cell-free biological therapeutic option in joint inflammations. It is necessary to continue the study of MPs in the context of inflammatory joint diseases to determine their value as biomarkers for diagnostic, prognostic, and therapeutic purposes.

## Figures and Tables

**Figure 1 ijms-20-05453-f001:**
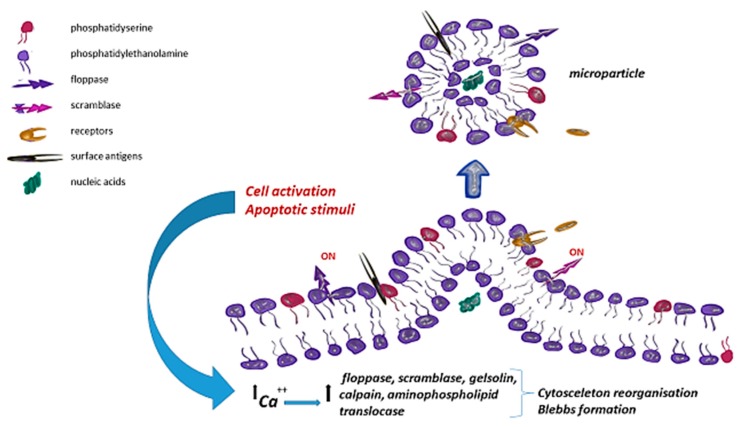
Microparicle formation following cellular activation and cytoskeletal.

**Table 1 ijms-20-05453-t001:** Cells of origin of microparticles and their clusters of differentiation.

Parent Cells	Surface Membrane Antigens of MPs Reflecting Their Cell of Origin
Platelets	CD41, CD41a, CD42a, CD42b, CD61, CD62p, PS, TF
Endothelium cells	CD31, CD51, CD62e, CD105, CD144, CD146, PS, TF
Erythrocytes	CD235a
Leukocytes	CD45
Monocytes	CD14, PS, TF
Neutrophils	CD66b
Th-cells	CD4
Ts-cells	CD8
B-cells	CD20

CD—cluster of differentiation, MPs—microparticles, PS—phosphatidylserine, TF—tissue factor.

**Table 2 ijms-20-05453-t002:** Microparticles in blood and other biological fluids in inflammatory joint diseases.

Disease	Microparticles	References
RA	Increased number of PMPs in peripheral blood and synovial fluid in RA	[8,49,50,51]
RA	Increased number of circulating MPs exposing complement components in early RA	[51]
RA	Increased number of monocyte-, B-cell-, T-cell-, platelet-derived MPs in high disease activity in RA	[51,52]
RA	Monocyte-derived MPs present in a much larger amount in synovial fluid than in plasma in RA	[53]
RA	Significantly increased number of granulocyte-derived MPs in synovial fluid in the RA patients with aCCP antibodies	[54]
RA	Increased number of MPs with CD3, CD14, and CD19 antigens in the urine of RA patients with high disease activity	[52]
JIA	Increased number of PMPs in synovial fluid in JIA compared to osteoarthritis	[8]
JIA	Much higher number of PMPs in synovial fluid in active JIA than in serum	[55]
PsA	Increased number of circulating PMPs and EMPs in PsA	[47]
PsA	Increased number of PMPs in synovial fluid in PsA	[8]
AS	Decrease in the number of MPMs and EMPs during the anti-TNFα treatment in AS	[56]
AS	No differences in the number of MPs between AS patients and healthy control, but significantly higher expression of CD4, CD62, CD14 and lower expression of CD41 in the MPs surface in AS	[57]

aCCP—anti cyclic citrullinated peptide antibodies; AS—ankylosing spondylitis; CD—cluster of differentiation; MPMs—monocyte-derived microparticles; PMPs—platelet-derived microparticles; PsA—psoriatic arthritis; RA—rheumatoid arthritis; JIA—juvenile idiopathic arthritis.

**Table 3 ijms-20-05453-t003:** Potential role of microparticles in the development of rheumatoid arthritis.

MPs as a Potential Pathogenetic Factor of RA	References
**Activation of Immunocompetent Cells**	
Activation of B-cells by macrophage/monocyte-derived MPs from synovial fluid	Messer et al. [59]
**Participation in Formation of Immune Complexes**	
Increased number of C1q, C4, C3-binding MPs in synovial fluid and in peripheral blood	Biro et al. [60]
mpIC present in synovial fluid	Cloutier et al. [61]
**Increased Secretion of Matrix Metaloproteinases**	
Monocyte- and B-cell-derived MPs can induce the release of MMP3, MMP9, MMP13 in FLS	Distler et al. [62]
**Modulation of Chemokine and Cytokine Release**	
Monocyte- and granulocyte-derived MPs from synovial fluid modulate MCP-1, IL-6, IL-8, and CCL5 release by synoviocytes	Berckmans et al. [63]
Increased secretion of TNFα and IL-1, IL-17 by monocytes stimulated by monocyte-, B-cell-, T-cell-, platelet-derived MPs from peripheral blood	Viñuela-Berni et al. [52]
**Pro-Coagulation Activity**	
Monocyte- and granulocyte-derived MPs from synovial fluid are strongly coagulant via the factor VII-dependent pathway	Berckmans et al. [64]
**Activation of Vascular Endothelium Cells**	
MPs from articular fluid stimulate FLS production and release of VEGF	Berckmans et al. [63]
Stimulating effect of leukocyte-derived MPs on production and release by rheumatoid synoviocytes of proangiogenic CXC	Reich et al. [65]

C—complement component; CCL—C-C motif chemokine ligand; CXC—CXC chemokines; FLS—fibroblast-like synoviocyte; IL—interleukin; MCP—monocyte chemoattractant protein; MMP—metalloproteinase; MPs—microparticles; RA—rheumatoid factor; TNF—tumor necrosis factor; VEGF—vascular endothelium growth factor.

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
