# Peer review of "Role of Microparticles in the Pathogenesis of Inflammatory Joint Diseases"

_ijms, 2019, doi:10.3390/ijms20215453_

Round 1

Reviewer 1 Report

This is a comprehensive review describing a very interesting and relevant topic.

Krajewska-Wloddarczyk et al. summarize current findings about microparticles in inflammatory joint diseases by describing roles of microparticles in different arthritic diseases, discussing there use as disease indicator and the role of mesenchymal stem-cell derived microparticles for therapy. In summary the review is very well written, I only have a few comments  to further improve the manuscript.

please include a list of abbreviations, as use of several abbreviations for different microparticles is sometimes confusing to the reader.

line 121: add reference

I would suggest adding a table to chapter “Role of microparticles in inflammatory joint diseases” to give a clearer summary of microparticles in different body fluids and different diseases.

In the summary authors suggest that MP determination can be used for disease activity assessment, while they stated before that microparticle detection is a technical challenge. Are further technical improvements necessary for the use of MP as indicators? And if so, can you please comment on this?

Author Response

Thank you very much for your valuable comments. The manuscript has been corrected according to all your comments:

please include a list of abbreviations, as use of several abbreviations for different microparticles is sometimes confusing to the reader.

- a list of abbreviation was included

line 121: add reference

- the reference was added

I would suggest adding a table to chapter “Role of microparticles in inflammatory joint diseases” to give a clearer summary of microparticles in different body fluids and different diseases.

- revised Table 2 was added

In the summary authors suggest that MP determination can be used for disease activity assessment, while they stated before that microparticle detection is a technical challenge. Are further technical improvements necessary for the use of MP as indicators? And if so, can you please comment on this?

We added more information to chapter "Methods of microparticle detection":

There are no standardized microparticle testing techniques and each laboratory conducting such testing develops its own MPs detection methods and standards, however common protocol among researchers is to start collecting MPs from blood with a centrifugation to collect platelet-free plasma to avoid activation and subsequent release of microparticles from platelets. Microparticle identification is a technical challenge because they are much smaller than cells of origin (diameters of 10–100× less). Microparticles can be isolated from blood, other biological fluids or from cell cultures. Since MPs are cell-derived structures a cytometric test is the “gold standard” and is the most widely used method of microparticle detection on the basis of light scattering as well as binding of marker antibodies to identify the cell of origin [34]. The MPs population to the flow cytometry settings is defined by using size calibration beads. However, MPs smaller than approximately 0.5 µm in diameter are not efficiently resolved by conventional flow cytometers. Available digital flow cytometers do not count all MPs because of their limited forward scatter (FS) sensitivity [35]. Recently, high-sensitivity flow cytometers with significantly improved light scatter detection became available and provide sufficient size resolution for the identification of MPs subtypes [36].

Thank you for your time and help in improving our manuscript

Reviewer 2 Report

This review article is of interest.  The authors report on the role of microparticles in the pathogenesis of 
 inflammatory joint diseases .

Major concerns

The abstract must be re-written to reflect the major findings/published data on the role of MPs in arthritides, rather than their reporting on their generic function, ways to measure them etc. This will help the Reader and will provide an outline on the experimental data so far provided.

Table 1 rather than explaining the cellular source of MPs gives a list of cells/cell subsets and their surface markers. The authors must provide a better text to help the Reader to understand that this table provides cells as a source of microparticles and surface markers specific for these cells which can be used for flowcytometry determination (of the cells? Their MPs? Both?) . I would like to know, for example, the best combination of surface combination to assess platelet-originate MPs.

The authors are in the unique position to provide a Table (revised Table 1?) giving a literature based estimation of MPs distribution  in relation to their cell origin in healthy and pathological conditions?

This will help us to understand that MPs are mostly and by far originating by platelets and megacaryocytes (90%?) and attempts to measure them must be focused in these cells rather than in other cell subsets except if…..?

Given that the literature on the role of MPs in joint disease is rather limited and is thoroughly discussed by the Authors, I would suggest to them that they considerably expand on MPs formation and methods of measurement/assessment. The authors are fully aware that this is a Journal of Molecular Sciences and I am pretty certain they can properly do that.

Figure 1 is a pretty one but legend to the Figure is poor. Please expand and give more details on microparticle formation as per legend/schematic representation.

The authors have the difficult task to expand on a topic (the pathogenic role of MPs in inflammatory joint diseases); this is a difficult task for a profound reason; little is known/studied in joint diseases.

Though the data are limited, I would suggest that they divide their text on sections based on disease entities (RA, Ps, etc) and within these sections provide experimental data on pathogenesis (animal studies, human studies, studies on pathogenesis, clinical relevance etc).

They also must expend a bit and give (at least in the form of a representative example) on an autoimmune disease entity whereby MPs have extensively been studied. This will assist the Reader to understand what needs to be done (experimental and translational research) in the near future in joint diseases. They may wish to refer to, for example. Systemic lupus erythematosus as a ‘ideal’ model or a choice of their own.

Author Response

Thank you very much for your valuable comments.
The manuscript has been corrected according to all your comments:

The abstract must be re-written to reflect the major findings/published data on the role of MPs in arthritides, rather than their reporting on their generic function, ways to measure them etc. This will help the Reader and will provide an outline on the experimental data so far provided.

the abstract was rewritten

Table 1 rather than explaining the cellular source of MPs gives a list of cells/cell subsets and their surface markers. The authors must provide a better text to help the Reader to understand that this table provides cells as a source of microparticles and surface markers specific for these cells which can be used for flowcytometry determination (of the cells? Their MPs? Both?) . I would like to know, for example, the best combination of surface combination to assess platelet-originate MPs.

more information were added to chapter "Methods of microparticle detection" Table 1 was revised

The authors are in the unique position to provide a Table (revised Table 1?) giving a literature based estimation of MPs distribution in relation to their cell origin in healthy and pathological conditions?

This will help us to understand that MPs are mostly and by far originating by platelets and megacaryocytes (90%?) and attempts to measure them must be focused in these cells rather than in other cell subsets except if…..?

Table 2 was revised including a literature of MPs distribution in relation to their cell origin. Circulating MPs are mostly platelet-derived, however it is very likely that the activation of monocytes/macrophages, B-cells, T-cells and endothelial cells observed in patients with inflammatory diseases may result in an increased release of MPs from these cells, raising their levels in plasma

Given that the literature on the role of MPs in joint disease is rather limited and is thoroughly discussed by the Authors, I would suggest to them that they considerably expand on MPs formation and methods of measurement/assessment. The authors are fully aware that this is a Journal of Molecular Sciences and I am pretty certain they can properly do that.

The chapter "Methods of microparticle detection" was rewritten

Figure 1 is a pretty one but legend to the Figure is poor. Please expand and give more details on microparticle formation as per legend/schematic representation.

More details on microparticle formation were added as the legend to the Figure 1

The authors have the difficult task to expand on a topic (the pathogenic role of MPs in inflammatory joint diseases); this is a difficult task for a profound reason; little is known/studied in joint diseases.

Though the data are limited, I would suggest that they divide their text on sections based on disease entities (RA, Ps, etc) and within these sections provide experimental data on pathogenesis (animal studies, human studies, studies on pathogenesis, clinical relevance etc).

The text was didided on recommended sections

They also must expend a bit and give (at least in the form of a representative example) on an autoimmune disease entity whereby MPs have extensively been studied. This will assist the Reader to understand what needs to be done (experimental and translational research) in the near future in joint diseases. They may wish to refer to, for example. Systemic lupus erythematosus as a ‘ideal’ model or a choice of their own.

Data on the specific role of microparticles in lupus pathogenesis was added to the chapter "Introduction"

Thank you Very much for your time and help in improving our manuscript.

Reviewer 3 Report

This manuscript review the possible function of MPs in osteoarthritis. The manuscript is well organized and logically summarize the importance of MPs. There are some points which is not completely supported by references. In " Methods of microparticle detection" Annexin V is used for detection of MPs but it is widely discussed that Annexin V is used to detect the early apoptotic processes of various cells or even chondrogenic cell cultures. It seems to be a brave statement that Annexin V can be used only for MP detection. This chapter has to be rewritten. In Table 2, the reviewer does not understand its significance and it is not informative enough.

Although the functions of MMPs are discussed briefly but no data can be found about aggrecenase or hyaluronidase functions. These matrix degrading enzymes supposed to be discussed.

Author Response

Thank you very much for your valuable comments.
The manuscript has been corrected according to your comments

There are some points which is not completely supported by references. In " Methods of microparticle detection" Annexin V is used for detection of MPs but it is widely discussed that Annexin V is used to detect the early apoptotic processes of various cells or even chondrogenic cell cultures. It seems to be a brave statement that Annexin V can be used only for MP detection. This chapter has to be rewritten.

- the chapter "Methods of microparticle detection" was rewritten. More explanations were added to the method: we do realize that MPs in flow cytometry are commonly determined with annexin V as other cell-derived membrane structures and then defining by their size within the interval of 0.1 to 1 μm- the main defining criterion. 

In Table 2, the reviewer does not understand its significance and it is not informative enough.

- Table 2 was revised

Although the functions of MMPs are discussed briefly but no data can be found about aggrecenase or hyaluronidase functions. These matrix degrading enzymes supposed to be discussed.

-There is no published data on the role of microparticles to stimulate  activity of aggrecenase or hyaluronidase in inflammatory join diseases. 

Thank you for your time and help in improving our manuscript